# Clinical Nomogram Model for Pre-Operative Prediction of Microvascular Invasion of Hepatocellular Carcinoma before Hepatectomy

**DOI:** 10.3390/medicina60091410

**Published:** 2024-08-28

**Authors:** Jen-Lung Chen, Yaw-Sen Chen, Kun-Chou Hsieh, Hui-Ming Lee, Chung-Yen Chen, Jian-Han Chen, Chao-Ming Hung, Chao-Tien Hsu, Ya-Ling Huang, Chen-Guo Ker

**Affiliations:** 1Department of General Surgery, E-Da Hospital, I-Shou University, Kaohsiung 824, Taiwan; sardo0926@gmail.com (J.-L.C.); ed102489@edah.org.tw (Y.-S.C.); ed105982@edah.org.tw (K.-C.H.); ed106128@edah.org.tw (H.-M.L.); ed104874@edah.org.tw (C.-Y.C.); ed109516@edah.org.tw (J.-H.C.); 2Department of General Surgery, E-Da Cancer Hospital, I-Shou University, Kaohsiung 824, Taiwan; ed100647@edah.org.tw; 3Department of Pathology, E-Da Hospital, I-Shou University, Kaohsiung 824, Taiwan; ed103797@edah.org.tw; 4Cancer Registration Center, E-Da Cancer Hospital, I-Shou University, Kaohsiung 824, Taiwan; ed100884@edah.org.tw

**Keywords:** hepatocellular carcinoma, microvascular invasion, recurrence, outcome, nomogram

## Abstract

*Background and Objectives*: Microvascular invasion (MVI) significantly impacts recurrence and survival rates after liver resection in hepatocellular carcinoma (HCC). Pre-operative prediction of MVI is crucial in determining the treatment strategy. This study aims to develop a nomogram model to predict the probability of MVI based on clinical features in HCC patients. *Materials and Methods*: A total of 489 patients with a pathological diagnosis of HCC were enrolled from our hospital. Those registered from 2012–2015 formed the derivation cohort, and those from 2016–2019 formed the validation cohort for pre-operative prediction of MVI. A nomogram model for prediction was created using a regression model, with risk factors derived from clinical and tumor-related features before surgery. *Results*: Using the nomogram model to predict the odds ratio of MVI before hepatectomy, the AFP, platelet count, GOT/GPT ratio, albumin–alkaline phosphatase ratio, ALBI score, and GNRI were identified as significant variables for predicting MVI. The Youden index scores for each risk variable were 0.287, 0.276, 0.196, 0.185, 0.115, and 0.112, respectively, for the AFP, platelet count, GOT/GPT ratio, AAR, ALBI, and GNRI. The maximum value of the total nomogram scores was 220. An increase in the number of nomogram points indicated a higher probability of MVI occurrence. The accuracy rates ranged from 55.9% to 64.4%, and precision rates ranged from 54.3% to 68.2%. Overall survival rates were 97.6%, 83.4%, and 73.9% for MVI(−) and 80.0%, 71.8%, and 41.2% for MVI(+) (*p* < 0.001). The prognostic effects of MVI(+) on tumor-free survival and overall survival were poor in both the derivation and validation cohorts. *Conclusions*: Our nomogram model, which integrates clinical factors, showed reliable calibration for predicting MVI and provides a useful tool enabling surgeons to estimate the probability of MVI before resection. Consequently, surgical strategies and post-operative care programs can be adapted to improve the prognosis of HCC patients where possible.

## 1. Introduction

Hepatocellular carcinoma (HCC) is a common disease and was ranked the second leading cause of cancer-related deaths in man and the fourth in women in Taiwan in 2019 [1]. Despite recent improvement in treatment strategies for HCC, high recurrence rates and unsatisfactory long-term survival after curative resection are prevalent, indicating that this is a life-threatening condition for most patients [2,3]. Furthermore, the global burden of HCC remains high, particularly in Asia and sub-Saharan Africa, due to the high prevalence of chronic hepatitis B in these regions [4]. Liver cancer is the third most common cause of cancer-related deaths worldwide, with HCC comprising the majority of liver cancer cases. The incidence rate of HCC decreased from 57.77 to 44.95 in 100,000 from 2013 to 2021 in Taiwan [5]. Although incidence rates have stabilized in recent years due to effective viral hepatitis control program and improved outcomes from early detection and advanced treatment, the burden of HCC is anticipated to rise again due to increasing survival rates and risk factor-related diseases [6]. The mortality rates for liver cancer have plateaued in recent years, but they remain high, particularly among certain racial and ethnic populations in the USA [7].

There is no specific diagnosis tool or method for pre-operative prediction of microvascular invasion(MVI) in HCC patients. Although several traditional staging systems exist, no single one is uniformly accepted [8,9]. Moreover, the criteria vary widely and cannot accurately predict individualized MVI and outcomes after surgery in HCC patients with different tumor burden [10]. The incidence of MVI may vary due to the different tumor stages [8,11,12]. For instance, positive MVI were found 18.1% patients (83 in 458 with single small HCC ≤ 3 cm) in a Chinese cohort [8], 29.55% of early HCC patients (13 in 44) in a France study [13], and 40.6% patients (168 in 414 early and single nodule ≤ 3 cm) in a Japanese study, respectively [12]. Emerging predictors of MVI, can be identified from the large-scale clinical data, including serum laboratory tests, histopathological patterns, radiomics, and genomics, as recommended in previous study models [11,14,15,16,17]. The current application models include clinical scoring of specific indicators through multivariate logistic analysis [14,16], novel nomogram [18], machine learning and artificial intelligence approaches [19,20].

The recurrence rates of HCC after non-surgical therapy or surgical resection can be as high as 70% within five years, depending on the patient’s disease stage, tumor pathology, and treatment strategy [3,21,22,23]. HCC patients with MVI have a 4.4-fold increased risk of tumor recurrence [24]. MVI plays a critical role in predicting postoperative recurrence and outcome [25,26]. To achieve a better prognosis, the treatment strategy should be based on the probability of MVI before surgical intervention [19,26]. A study of independent predictors for MVI identified three variates; tumor diameter, alpha-fetoprotein (AFP) and des-gamma-carboxy prothrombin [27]. Matching among the three predictors can predict MVI positivity in such patients, suggesting that anatomic or more extended liver resection may lead to better survival [12]. According to the study of Wang et al, the MVI scoring system can determine the prognosis of HCC patients after curative liver resection and guide the selection of patients with high risk of recurrence to receive adjuvant therapy for the prevention of recurrence [26]. Therefore, the efficacy of pre-operative prediction of MVI informed by a therapeutic strategy results in a better prognosis.

Previous studies have clearly demonstrated that the tumor’s pathological MVI is notably associated with tumor recurrence [28]. Additionally, MVI is recognized as an independent predictor of overall and disease-free survival among HCC patients [29]. In order to optimize the therapeutic decision-making for HCC patients, the pre-operative prediction of MVI is crucial. In case of high MVI probability, the patient should be considered for extended anatomic liver resection or revising surgical strategies following the guidelines without delay [30]. Thus, this study has two objectives: to predict the probability of MVI before operation using the clinical nomogram model, and to evaluate the effects of MVI on the prognosis of HCC after hepatectomy.

## 2. Patients and Method

### 2.1. Study Design

All patients were retrospectively collected from the cancer registry information system database of E-Da hospital. The criteria included adult patients receiving hepatectomy as a curative treatment and with a pathological diagnosis of HCC. Patients treated from 2012 to 2015 were used as the derivation cohort (D-cohort), and those from 2016 to 2019 as the validation cohort (V-cohort) for the pre-operative prediction of MVI probability. Patients were sub-grouped as MVI-negative (MVI(−)) or MVI-positive (MVI(+)) based on the pathological findings. The inclusion criteria were (a) patients with imaging and a pathological diagnosis of HCC, (b) patients treated with surgical liver resection, (c) adults age ≥ 18 years old. The exclusion criteria included (a) patient with severe systemic diseases, and (b) those who were deceased from non-HCC factors within the two months postoperatively. Two endpoints of this study were tumor-free survival (TFS) and overall survival (OS), calculated from the date of diagnosis. The study was approved by the Institutional Review Board and Ethical Committee of E-Da Hospital (code no. EDAHS 110012) on 1 September 2022.

### 2.2. Pre-Operative Patient Variates Profiles

Basic profiles of patients included their age, gender, history of alcohol use, diabetics (DM), BMI, hepatitis B/C, TMN stage, Child–Pugh classification, Barcelona Clinic Liver Cancer (BCLA) stage, and tumor burden, etc., shown in Table 1. Pre-operative laboratory profiles included WBC count (10^3^/µL) and their classification of neutrophil, monocyte, and lymphocyte, as well as platelet (10^3^/µL). Chemistry serum study included Hemoglobin A1c (HbA1c, %), International Normalized Ratio for prothrombin time(INR), Indocyanine green test (ICG, %), alfa-fetoprotein (AFP, ng/mL), bilirubin (Bil, mg/dL), glutamic oxaloacetic transaminase (GOT, µ/L)/glutamic pyruvate transaminase (GPT, µ/L) ratio, alkaline phosphatase (Alk-p, µ/L) and albumin (Alb. g/dL levels. Combination nutritional indices were examined, namely the prognostic nutritional index (PNI) [31], albumin-to-alkaline phosphatase ratio (AAR) [32], albumin-bilirubin score (ALBI) [33], and geriatric nutritional risk index(GNRI) [34], as shown in Table 2. All variables were stratified and selected for evaluation if their prediction power was significant for establishing a nomogram for prediction of MVI probability.

### 2.3. Statistical Analysis

Statistical analysis was performed using SPSS (Version 25.0, IBM) and R statistical software (Version 3.3.3, https://www.r-project.org and accessed on 28 February 2024, RStudio 2023.12.0+369 for Windows). Multiple variates were initially selected to identify the most relevant features. Variables not meeting the consistency threshold or showing no statistical differences (*p* > 0.05) were removed based on the Student’s *t*-test. As process preconditions, the predictive model was used to obtain the most relevant features. Univariate analysis was first used to compare the clinical variables between the two groups; MVI(−) or MVI(+), using the chi-square test and Wald test score to categorize the variables and independent Student *t*-test for continuous variables where appropriate. Additionally, logistic regression model analysis for odds ratio, and receiver operating characteristic (ROC) curves for sensitivity and specificity were utilized. The best cutoff values of these variables were determined via the maximum Youden index, generated from the respective ROC curves, to evaluate the predictive values for MVI. Nomogram construction was performed and established using multivariate analysis of patients’ data. The included variables were selected using stepwise multiple logistic regression analysis. Survival curves were constructed and compared using the Kaplan–Meier survival method. Tumor-free survival(TFR), and overall survival(OS) rates at 1, 3, 5 years were assessed for surgical outcomes. The concordance index (C-index), ROC and area under the curve (AUC) were used to assess discriminatory probability. The calibration probability was assessed using calibration curves and considered to have predictive power. Differences in AUC values between these models were analyzed using the Delong test. *p* < 0.05 was considered statistically significant.

## 3. Results

### 3.1. Demographic of Clinical Profiles of D-Cohort and V-Cohort

The clinical characteristics of 489 HCC patients in total (281 in D-cohort and 206 in V-cohort) were stratified by MVI(−) and MVI(+), as shown in Table 1. Odds ratio of MVI was obtained using univariates logistic regression analysis from the basic and tumor profiles, as shown in Appendix A. Significant pre-operative variables were identified first, including HbA1c, BCLC stage, tumor extension, satellite nodule, tumor size, and tumor number.

### 3.2. Clinical Laboratory Data and Nutrition-Based Index for HCC Patients

Clinical laboratory data and nutrition-based indices of HCC patients in the D-cohort and V-cohort were also stratified by MVI(−) and MVI(+), as shown in Table 2. The odd ratio of MVI was obtained using univariate logistic regression analysis based on laboratory data and nutrition-based indices. Significant pre-operative variables were identified, including AFP, bilirubin platelet, GOT/TPT ratio, Alk-p, AAR, ALBI and GNRI, as shown in the Appendix A. All significant variables from Appendix A were used for multiple logistic regression analysis. Finally, the significant predictive risk factors of MVI determined through the univariate and multiple logistic regression analysis are shown in Table 3. Using the random forest method to predict odds ratio of MVI before hepatectomy for D-cohort, we identified and assessed the AFP, platelet, GOT/GPT ratio, AAR, ALBI and GNRI as significant variates for MVI prediction, as listed in Table 4. The forest plot diagram is demonstrated in Figure 1.

### 3.3. Calibration Plot Model and Nomogram for Prediction Probability

The calibration plot model diagram was constructed using the significant variables, including the AFP, platelet, GOT/GPT ratio, AAR, ALBI and GNRI, to predict the odds ratio of MVI before hepatectomy (Figure 1). A nomogram was constructed from seven valuables to predict MVI risk in HCC patients pre-operatively. To use this nomogram, scores for each variable were obtained after logarithmic transformation on the corresponding axis. A line drawn from the total points’ axis to the risk and determination of the probability of MVI risk with a C-index score was 0.756, as in Figure 2. The evaluation of each valuable risk factor included threshold, sensitivity, specificity, Youden index, accuracy precision and AUC score, as listed in Table 4. The Youden index scores for prediction risk variables were 0.287, 0.276, 0.196, 0.185, 0.115, 0.112, and 0.135 for the AFP, platelet, GOT/GPT ratio, AAR, ALBI and GNRI, respectively. Total maximal score from the nomogram was determined to be 220 points. Increasing risk scores in the nomogram increased the prediction probability of MVI occurrence.

### 3.4. Effect of MVI on Tumor-Free Survival and Overall Survival Rates after Hepatectomy

The Kaplan–Meier method was used to estimate the survival rate, and the log-rank test was used to evaluate the difference between the two groups. In the D-cohort, the tumor-free survival (TFS) rates of MVI(−) were 87.2%, 59.2%, and 47.9%, and of MVI(+) 65.0%, 47.4% and 45.2% for 1, 3, and 5 years, respectively (*p* = 0.14). The overall survival rates (OS) of MVI(−) were 97.6%, 83.4%, and 73.9%, and of MVI(+) 80.0%, 71.8% and 41.2% for 1, 3, and 5 years, respectively (*p* ≤ 0.001) (Figure 3A,B). In the D-cohort, the TFS rates of MVI(−) were 95.5%, 76.8%, and 71.5%, and of MVI(+) 84.0%, 63.3%, and 44.9% for 1, 3, and 5 years, respectively (*p* = 0.001) (Figure 3A,B). The OS rates of MVI(−) were 91.5%, 72.9%, and 66.4%, and of MVI(+) 78.2%, 57.0%, and 53.4% for 1, 3, and 5 years, respectively (*p* = 0.021) (Figure 4A,B). The effects of MVI(+) on the TFS and OS were poor in both the D-cohort and V-cohort.

### 3.5. ROC and AUC Score from Multiple Logistic Regression Model in Predicting MVI Factors

The ROC and AUC were used to assess discriminatory probability. The calibration probability was assessed using calibration curves to evaluate predictive power. The ROC curve and AUC score were obtained for the significant risk factors from multiple logistic regression model in predicting MVI positivity. The seven significant variables of AUC for predicting were 0.684, 0.631, 0.627, 0.550, 0.564, 0.521 and 0.521 for the tumor size, AFP, platelet, GOT/GPT ratio, AAR, ALBI, and GNRI, respectively, as shown in Figure 5. Therefore, AAR and GNRI were considered less predictive of MVI risk. The other five variables increased the probability of pre-operative MVI prediction.

## 4. Discussion

The notable pathological findings of MVI include degradation of the basal membrane and herniation of the tumor cells to the capillary lumen. During this process, tumor clusters are coated with endothelial cells, forming micro-emboli within the capillary lumen. Based on the hepatocarcinogenesis, tumor-induced neo-angiogenesis, proliferation, and inhibition of apoptosis constitute the backbone of MVI [35]. Additionally, MVI usually exhibits poor differentiation, programmed death ligand-1 expression, and frequently tumor nests surrounded by blood vessels [36]. Pathological finding of MVI can serve as a source for spreading tumor circulating cells and metastasis. Type of vessel invaded determines the nature of the intrahepatic or extrahepatic spread of HCC tumor cells. The invasion of the portal vein as an efferent vessel leads to intrahepatic spread, whereas invasion of hepatic venous tributaries leads to systemic metastasis, especially lung metastasis [36,37]. Histopathological identified MVI was found in 211 of 707 patients (29.8%) and 89 of 297 patients (30.0%) in the training and validation cohorts, respectively, in a study reported from Shanghai, China [38]. In our current study, MVI was found in 26.7% of the D-cohort and 34.6% of the V-cohort. The detection of MVI in HCC histopathological evaluation and prediction indicates a higher risk of disease dissemination and a worse prognosis. Therefore, MVI should be predicted pre-operatively to deliver appropriate pre-/post-operative treatment options to patients.

MVI is an expression of aggressive biological behavior of the tumor. If MVI is presented, it is recognized as one of the most critical factors predictive of HCC early recurrence and poor prognosis. Therefore, it is essential to assess the probability of MVI before operation. Various studies have attempted to provide prospective methods to measure MVI in HCC. Routine serologic laboratory tests could provide convenient biomarkers for the prediction of MVI when possible. For constructing a nomogram, associations among the clinical serological profiles, tumor burden and inflammation indexes were used for assessment probability of MVI using logistic regression analysis before operation. Multivariate analysis showed that specific risk variables were predictors for MVI. The nomogram performed well in terms of its discrimination ability and actual observation in both cohorts. However, the selected risk variables differ in each study based on the analysis models. Wang et al. demonstrated that risk variables included the serum AFP (OR = 1.117), tumor size (OR = 1.005) and tumor number (OR = 1.101) by nomogram model in Singapore [39]. Another study reported that positive variables included the AFP, PIVIK-II and tumor size, which were used for prediction of MVI through a machine learning model in China [19]. Most selected variables used for scoring are not unique but only AFP was consistently used in most studies. High levels of AFP have been proven to correlate with more aggressive behaviors [18,40,41,42,43]. In our current study, peripheral platelets served as a positive predictor for MVI. Previous studies have also shown that platelets participate in multiple steps of tumorigenesis, including tumor growth, extravasation and metastasis [44]. In addition to serological laboratory data, other common tools for pre-operative prediction of MVI were imaging techniques, including CT scanning or MRI [45,46]. Recently, peripheral circulatory tumor cells and genomics have also been used for prediction, although these methods are time-consuming and expensive [47,48].

MVI of HCC is a common pathological finding and is associated with early tumor recurrence with a reduced survival rate [49]. MVI occurs from 15% to 60% of resected HCC specimens [8,12,13,38]. The accurate pre-operative prediction of MVI can assist surgeons in choosing a better surgical procedure initially. In cases where pre-operative probability of MVI is determined, a treatment strategy must be chosen accordingly. This might more often include the extension of resection, pre-operative regional ablation or neo-adjuvant therapy before definite liver resection, in line with the BCLC criteria of the 2022 updated edition for liver resection [50]. Fundamentally, MVI sites are usually found near the tumor margin, with closer position being the most common. Pathologically, MVI has been observed in 79.2% of cases within 10 mm between the surgical margin and primary lesion in HCC patients with tumor size less than 3 cm. These results suggest that a 10 mm resection margin is essential to sufficiently eradicate MVI surrounding the primary tumor, and a wider surgical margin greater than 10 mm was proposed for HCC patients with MVI by Nakashima et al. [51] A resection margin less than 10 mm is a surgical predictor of early recurrence for the MVI(+) group (HR, 0.68; 95% CI, 0.54–0.87; *p* = 0.002), but not for the MVI(−) group [52]. Hence, a wide surgical margin significantly improves oncological outcomes and long-term survival in HCC patients with MVI. That is the reason why the pre-operative prediction of MVI is an essential contribution to precise surgical decision-making [53]. Additionally, anatomic resection is generally recommended and has been shown to improved overall survival compared to the non-anatomic resection of HCC [54]. In cases of HCC with MVI, a wide surgical margin is an independent risk factor compared to anatomic resection. A study by Liu et al. showed that, for patients with MVI-(+) HCC, non-anatomic resection with a wide margin was a protective factor for overall survival and the time to recurrence compared to anatomic resection with narrow margins (HR = 0.618 vs. 0.662) [55].

Post-operative sequential therapy for HCC patients with MVI can improve the outcome after liver resection. A systemic review and meta-analysis indicated that post-operative adjuvant trans-arterial chemoembolization (pa-TACE), radiotherapy, radiofrequency ablation (RFA), sorafenib and conservative treatment were used as subsequent therapies for HCC patients with MVI after resection. The pa-TACE significantly improved the overall survival and recurrence-free survival compared with postoperative conservative treatment after curative resection (HR: 0.64, *p* < 0.001 and HR: 0.70, *p* < 0.001, respectively). The pa-TACE, postoperative radiotherapy and sorafenib can improve the prognosis of HCC patients with MVI after curative resection [56]. In consideration of the high-risk factors for recurrence, especially in HCC patients with MVI, pa-TACE treatment has gradually become the recommended method for prophylactic treatment to prevent early recurrence [57].

Pre-operative prediction of MVI is crucial for determining treatment strategies in cases where MVI is presented before resection. Although we have developed a nomogram to predict the probability of MVI, it is imperative to validate its accuracy, reliability, and clinical outcomes in a validation cohort. In this study, the sample size was not large, representing a limitation. Besides, several factors, such as BCLC classification, number of tumor, or pathological grading, etc., could play a role as a risk for prediction, but these were without significance in this study. Moreover, further research using patient’s radiomics data from CT or MRI scans, constituting more precise information, is necessary to enhance the validity and clinical applicability of our nomogram model for the pre-operative prediction MVI in future.

## 5. Conclusions

The nomogram model integrating clinical features demonstrates strong discriminatory and calibration abilities for the prediction of MVI in HCC patients. We have established this predictive model based on AFP, platelet count, GOT/GPT ratio, AAR, ALBI, and GNRI to access the likelihood of MVI in HCC patients before hepatectomy. We anticipate that this model will serve as an effective tool for surgeons in predicting MVI and optimizing treatment strategies to improve long-term outcomes for HCC patients with MVI.

## Figures and Tables

**Figure 1 medicina-60-01410-f001:**
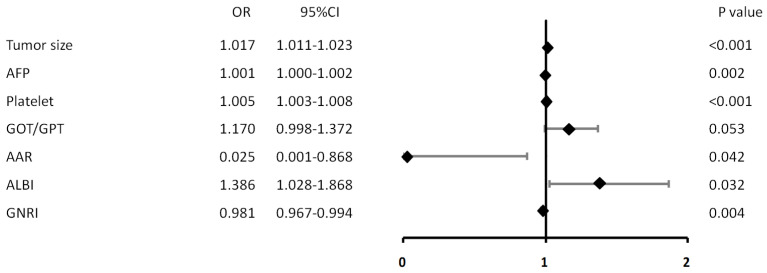
Significant risk factors in multiple logistic regression are presented as a calibration plot for prediction of the odds ratio of MVI.

**Figure 2 medicina-60-01410-f002:**
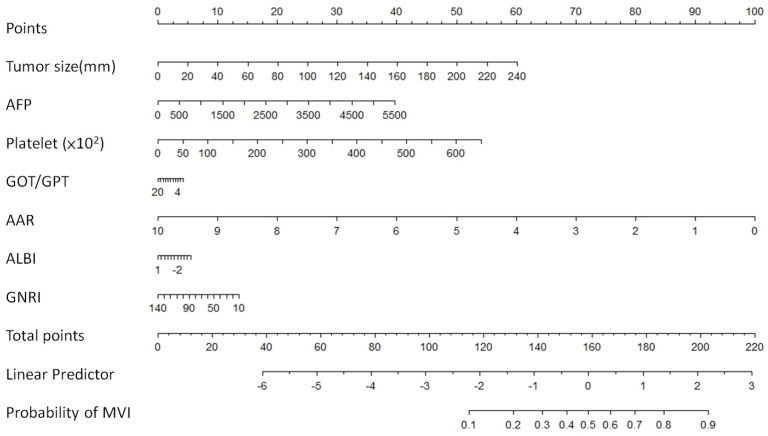
Significant predictive factors of the probability of MVI are presented in a nomogram for prediction of MVI in HCC patients before hepatectomy. Total number of points for all seven variables for the pre-operative prediction of MVI is 220 with a C-index of 0.756.

**Figure 3 medicina-60-01410-f003:**
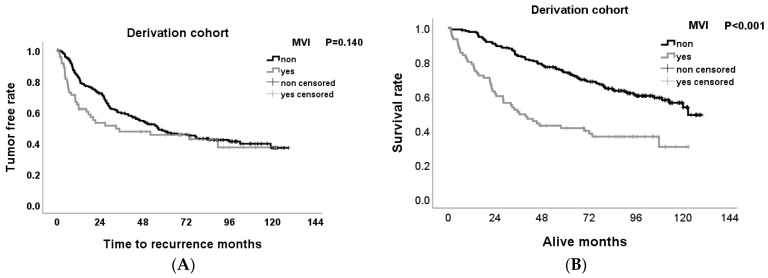
Tumor-free rates (**A**) and overall survival rates (**B**) of the patients with and without MVI after hepatectomy in the D-cohort.

**Figure 4 medicina-60-01410-f004:**
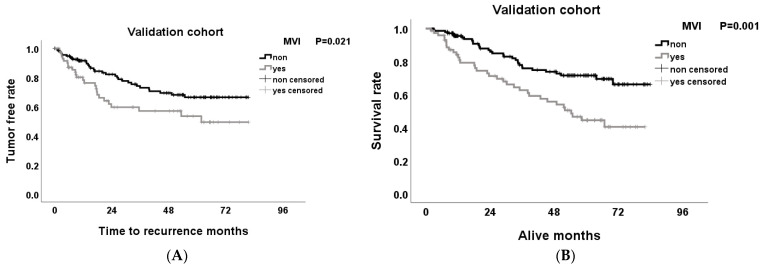
Tumor-free rates (**A**) and overall survival rates (**B**) of the patients with and without MVI after hepatectomy in the V-cohort.

**Figure 5 medicina-60-01410-f005:**
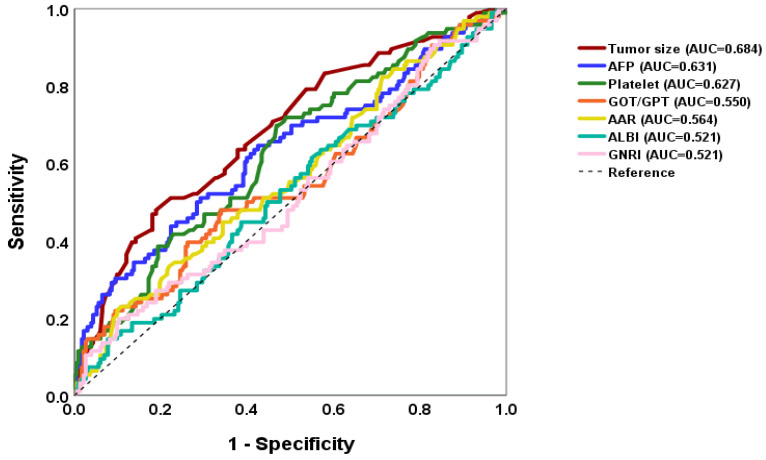
The ROC curve and AUC score of significant risk factors from a multiple logistic regression model for pre-operative prediction of MVI. The x-axis represents the model’s predicted probability of MVI (specificity), typically ranging from 0 to 1. This reflects the model’s estimate of the probability of MVI occurring. The y-axis represents the actual observed rate of MVI occurrence (sensitivity), also ranging from 0 to 1, indicating the proportion of MVI occurring in real data. The seven significant variables were the tumor size, AFP, platelet, GOT/GPT ratio, AAR, and GNRI. However, elevation of AAR and GNRI were associated with a reduced risk of MVI. The other five variables were used for pre-operative prediction of high probability of MVI.

**Table 1 medicina-60-01410-t001:** Profiles of patients of HCC of D-cohort and V-cohort after hepatectomy.

Variables	Total n = 489 n (%)	D-Cohort n = 281					V-Cohort n = 208				
		MVI(−) n = 206		MVI(+) n = 75		*p*	MVI(−) n = 136		MVI(+) n = 72		*p*
		n	%	n	%		n	%	n	%	
Age	60.8 ± 10.7	60.4 ± 10.9		60.8 ± 10.9		0.788	61.3 ± 9.6		60.8 ± 12.3		0.786
<65	307 (62.8)	135	73.4	49	26.6	0.975	80	65.0	43	35.0	0.900
≥65	182 (37.2)	71	73.2	26	26.8		56	65.9	29	34.1	
Sex											
Male	378 (77.3)	152	70.7	63	29.3	0.074	109	66.9	54	33.1	0.391
Female	111 (22.7)	54	81.8	12	18.2		27	60.0	18	40.0	
BMI	24.9 ± 3.8	25.1 ± 4.0		24.7 ± 3.4		0.522	25.1 ± 3.7		24.2 ± 3.8		0.077
<24	200 (41.5)	88	73.3	32	26.7	0.968	50	62.5	30	37.5	0.515
≥24	282 (58.5)	114	73.5	41	26.5		85	66.9	42	33.1	
DM											
yes	126 (25.8)	52	74.3	18	25.7	0.831	37	66.1	19	33.9	0.899
non	363 (74.2)	154	73.0	57	27.0		99	65.1	53	34.9	
HbA1c (%)	7.1 ± 1.6	7.6 ± 2.0		6.4 ± 1.5		0.048	7.1 ± 1.4		6.9 ± 1.3		0.581
<6.5	39 (41.1)	9	52.9	8	47.1	0.085	15	68.2	7	31.8	0.952
≥6.5	56 (58.9)	21	77.8	6	22.2		20	69.0	9	31.0	
Alcohol											
non	317 (66.3)	129	75.0	43	25.0	0.471	98	67.6	47	32.4	0.339
1 + 2	161 (33.7)	71	71.0	29	29.0		37	60.7	24	39.3	
ECOG											
0 + 1	482 (99.2)	203	73.8	72	26.2	1.000	136	65.7	71	34.3	0.168
2 + 3 + 4	4 (0.8)	2	66.7	1	33.3		0	0	1	100	
BCLC											
0 + 1	281 (57.5)	133	81.6	30	18.4	<0.001	90	76.3	28	23.7	<0.001
2 + 3	208 (42.5)	73	61.9	45	38.1		46	51.1	44	48.9	
Child–Pugh											
A	471 (98.3)	194	72.7	73	27.3	0.668	134	65.7	70	34.3	1.000
B	8 (1.7)	4	66.7	2	33.3		1	50.0	1	50.0	
Hepatitis											
non	93 (19.0)	38	71.7	15	28.3	0.768	30	75.0	10	25.0	0.155
B/C	396 (81.0)	168	73.7	60	26.3		106	63.1	62	36.9	
T location *											
seg. 1	7 (1.4)	4	100	0	0	0.462	3	100	0	0	0.323
left (seg. 2–3)	94 (19.3)	40	69.0	18	31.0		20	55.6	16	44.4	
right (seg. 4–8)	380 (77.9)	159	74.3	55	25.7		111	66.9	55	33.1	
seg. 2 + 3	7 (1.4)	3	60.0	2	40.0		1	50.0	1	50.0	
T extension											
non	451 (92.2)	194	76.1	61	23.9	0.001	132	67.3	64	32.7	0.026
yes	38 (7.8)	12	46.2	14	53.8		4	33.3	8	66.7	
T size (mm)	52.5 ± 36.4	45.5 ± 29.5		70.1 ± 38.7		<0.001	45.8 ± 32.9		67.2 ± 46.8		0.001
<50 mm	292 (59.7)	139	83.7	27	16.3	<0.001	92	73.0	34	27.0	0.004
≥50 mm	197 (40.3)	67	58.3	48	41.7		44	53.7	38	46.3	
T number	1.06 ± 0.41	1.06 ± 0.38		1.21 ± 0.60		0.039	1.00 ± 0.33		1.03 ± 0.34		0.471
1	421 (90.1)	179	76.8	54	23.2	0.002	124	66.0	64	34.0	0.322
≥2	46 (9.9)	19	52.8	17	47.2		5	50.0	5	50.0	
Satellite nodule											
no	415 (84.9)	189	81.8	42	18.2	<0.001	130	70.7	54	29.3	<0.001
yes	74 (15.1)	17	34.0	33	66.0		6	25.0	18	75.0	
Pre-op Tx **											
no	434 (88.8)	184	74.8	62	25.2	0.135	124	66.0	64	34.0	0.594
yes	55 (11.2)	22	62.9	13	37.1		12	60.0	8	40.0	

* T for tumor. seg. for segment; ** Pre-op Tx: pre-operative treatment with regional local ablation or trans-hepatic arterial chemo-embolization.

**Table 2 medicina-60-01410-t002:** Clinical laboratory data of HCC patients with/without MVI after hepatectomy.

Variables	Total n = 489 n (%)	D-Cohort n = 281					V-Cohort n = 208				
		Micro(−) n = 206		Micro(+) n = 75		*p*	Micro(−) n = 136		Micro(+) n = 72		*p*
		n	%	n	%		n	%	n	%	
INR	1.24 ± 1.82	1.12 ± 0.83		1.54 ± 3.38		0.283	1.20 ± 1.33		1.34 ± 2.31		0.598
<1.24	467 (95.5)	198	73.3	72	26.7	0.999	131	66.5	66	35.5	0.195
≥1.24	22 (4.5)	8	72.7	3	27.3		5	45.5	6	54.5	
ICG (%)	12.48 ± 9.23	12.78 ± 8.86		11.88 ± 6.48		0.422	12.54 ± 9.00		12.16 ± 12.59		0.802
<20	416 (87.0)	174	72.5	66	27.5	0.505	111	63.1	65	36.9	0.320
≥20	62 (13.0)	28	77.8	8	22.2		19	73.1	7	26.9	
AFP (×10^2^ ng/mL)	89.95 ± 446.61	31.66 ± 220.06		166.19 ± 449.3		0.017	44.36 ± 358.09		256.53 ± 841.3		0.045
<200 IU	355 (71.0)	150	79.8	38	20.2	<0.001	103	70.1	44	29.9	0.004
≥200IU	137 (29.0)	49	59.0	34	41.0		26	48.1	28	51.9	
WBC (×10^3^/µL)	6.58 ± 2.51	6.40 ± 2.67		6.87 ± 2.56		0.184	6.51 ± 2.15		6.96 ± 2.58		0.182
<5.93	213 (44.1)	102	78.5	28	21.5	0.087	61	73.5	22	26.5	0.048
≥5.93	270 (55.9)	102	69.4	45	30.6		74	60.2	49	39.8	
Neutro (×10^3^/µL)	4.59 ± 2.63	4.20 ± 2.71		4.94 ± 2.51		0.160	4.77 ± 2.36		4.99 ± 2.93		0.714
<7.65	176 (88.4)	80	73.4	29	26.6	0.007	45	67.2	22	32.8	0.186
≥7.65	23 (11.6)	4	33.3	8	66.7		5	45.5	6	54.5	
Lympho (×10^3^/µL)	1.71 ± 0.96	1.75 ± 1.12		1.75 ± 0.96		0.980	1.64 ± 0.69		1.65 ± 0.88		0.949
<2.38	164 (83.2)	71	71.0	29	29.0	0.331	44	68.8	20	31.3	0.200
≥2.38	33 (16.8)	12	60.0	8	40.0		6	46.2	7	53.8	
Platelet (×10^3^/µL)	194.02 ± 77.5	180.33 ± 67.4		205.93 ± 71.2		0.006	191.26 ± 66.9		226.42 ± 112.4		0.017
<178.50	218 (45.0)	109	82.6	23	17.4	0.001	64	74.4	22	25.6	0.026
≥178.50	266 (55.0)	95	65.5	50	34.5		72	59.5	49	40.5	
Bil, (mg/dL)	0.92 ± 0.37	0.95 ± 0.37		0.91 ± 0.39		0.437	0.87 ± 0.37		0.93 ± 0.35		0.275
<1.2	379 (79.0)	153	70.2	65	29.8	0.019	108	67.1	53	32.9	0.247
≥1.2	101 (21.0)	48	85.7	8	14.3		26	57.8	19	42.2	
GOT/GPT (µL)	1.35 ± 1.34	1.20 ± 0.76		1.51 ± 1.20		0.040	1.35 ± 1.81		1.60 ± 1.65		0.342
<1.35	338 (70.1)	152	78.8	41	21.2	0.002	102	70.3	43	29.7	0.023
≥1.35	144 (29.9)	49	60.5	32	39.5		34	54.0	29	46.0	
Alb. (g/dL)	4.14 ± 0.36	4.15 ± 0.40		4.12 ± 0.36		0.516	4.17 ± 0.29		4.09 ± 0.37		0.163
A < 4.15	213 (45.1)	88	71.0	36	29.0	0.201	53	56.9	36	40.4	0.136
≥4.15	259 (54.9)	112	77.8	32	22.2		80	69.6	35	30.4	
Alk-P (µL)	338.68 ± 277.1	321.03 ± 201.4		486.35 ± 605.5		0.069	305.85 ± 119.0		317.35 ± 142.7		0.594
A < 384.50	289 (42.6)	120	81.1	28	18.9	0.003	80	66.1	41	33.9	0.364
≥384.50	84 (57.4)	31	60.8	20	39.2		19	57.6	14	42.4	
PNI	41.42 ± 3.59	41.52 ± 3.96		41.17 ± 3.58		0.517	41.66 ± 2.88		40.95 ± 3.70		0.162
<41.00	165 (35.0)	71	71.7	28	28.3	0.402	38	57.6	28	42.4	0.114
≥41.00	307 (65.0)	129	76.3	40	23.7		95	68.8	43	31.2	
AAR	0.16 ± 0.70	0.17 ± 0.08		0.13 ± 0.07		0.014	0.16 ± 0.06		0.15 ± 0.06		0.819
<0.09	53 (15.3)	20	60.6	13	39.4	0.015	10	50.0	10	50.0	0.147
≥0.09	294 (84.7)	130	80.2	32	19.8		88	66.7	44	33.3	
ALBI	−2.67 ± 0.62	−2.70 ± 0.54		−2.49 ± 0.95		0.087	−2.74 ± 0.52		−2.66 ± 0.54		0.310
<−2.86	181 (37.7)	81	77.9	23	22.1	0.185	56	72.7	21	27.3	0.074
≥−2.86	299 (62.3)	120	70.6	50	29.4		78	60.5	51	39.5	
GNRI	106.8 ± 14.2	107.90 ± 12.8		101.88 ± 21.6		0.026	108.33 ± 11.1		105.93 ± 12.5		0.157
<98.87	77 (15.9)	33	63.5	19	36.5	0.076	7	28.0	18	72.0	<0.001
≥98.87	407 (84.1)	170	75.6	55	24.4		128	70.3	54	29.7	

**Table 3 medicina-60-01410-t003:** Predictive risk factor of MVI resulting from univariate and multiple logistic regression analysis. (Variates selected from Appendix A).

Variables	Univariate		Multivariate	
	OR (95% CI)	*p*	OR (95% CI)	*p*
BCLC				
0 + 1(ref.)	1		1	
2 + 3	2.876 (1.930, 4.284)	<0.001	1.119 (0.510, 2.458)	0.779
Tumor extension				
non (ref.)	1		1	
yes	3.586 (1.824, 7.051)	<0.001	1.771 (0.575, 5.453)	0.319
Tumor size (mm)	1.017 (1.011, 1.023)	<0.001	1.006 (0.995, 1.017)	0.306
Tumor number	1.695 (1.062, 2.705)	0.027	1.236 (0.616, 2.482)	0.551
Satellite nodule				
non (ref.)	1		1	
yes	7.368 (4.283, 12.677)	<0.001	5.660 (2.553, 12.547)	<0.001
AFP (×10^2^ ng/mL)	1.001 (1.000, 1.002)	0.002	1.000 (1.000, 1.001)	0.200
Platelet (×10^3^/µL)	1.005 (1.003, 1.008)	<0.001	1.004 (1.000, 1.008)	0.031
Alk.P (µL)	1.001 (1.000, 1.002)	0.033	1.000 (0.999, 1.001)	0.810
AAR	0.025 (0.001, 0.868)	0.042	2.674 (0.016, 450.862)	0.707
ALBI	1.386 (1.028, 1.868)	0.032	0.915 (0.303, 2.766)	0.875
GNRI	0.981 (0.967, 0.994)	0.004	0.997 (0.962, 1.034)	0.878

**Table 4 medicina-60-01410-t004:** Significant variables for MVI prediction after logistic regression analysis.

Pre-Operative Risk Variable	Threshold	Sensitivity	Specificity	Youden Index	Accuracy	Precision	AUC
Tumor size (mm)	42.50	66.7%	62.0%	0.287	64.4%	63.7%	0.684
AFP(×10^2^ ng/mL)	30.24	66.0%	61.6%	0.276	63.8%	63.2%	0.631
Platelet(×10^3^/µL)	178.50	68.8%	50.9%	0.196	59.9%	58.4%	0.627
GOT/GPT	1.33	43.4%	75.1%	0.185	59.3%	63.5%	0.550
AAR	0.09	23.2%	88.3%	0.115	55.8%	66.5%	0.564
ALBI	−2.86	70.3%	40.9%	0.112	55.6%	54.3%	0.521
GNRI	98.87	25.3%	88.2%	0.135	56.8%	68.2%	0.521

## Data Availability

The original contributions presented in the study are included in the article, further inquiries can be directed to the corresponding author.

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
