# Peer review of "Clinical Nomogram Model for Pre-Operative Prediction of Microvascular Invasion of Hepatocellular Carcinoma before Hepatectomy"

_medicina, 2024, doi:10.3390/medicina60091410_

Round 1
Reviewer 1 Report
Comments and Suggestions for Authors
In this paper, the authors demonstrate the evaluation of a model for prediction of Microvascular Invasion of Hepatocellular Carcinoma per-transplantation. I felt the paper made good results. However, It would benefit from rewording by an expert English editor to make the text clearer and more understandable. Some phrases are vague and difficult to understand. Below are outlined specific issues that should be addressed:
· In the introduction section, please clirifeis abbervaiation in advance.
· Please clarifies the number of patients which are involed in study
· Please describe the inclusion criteria and exculsion criteria of included patients.
· Please in the reporting of experimental measurements and of derived quantities, use internationally approved nomenclature, symbols, units, and standards.
· Result section: Please focus on presenting your results with specific details.
· Conclusion section: It would be beneficial to rephrase the conclusion with more comprehensive sentences that effectively summarize the key findings and implications of the study. Please consider revising the conclusion to ensure a more concise and impactful summary of the research.
Comments on the Quality of English LanguageIt would benefit from rewording by an expert English editor to make the text clearer and more understandable. Some phrases are vague and difficult to understand
Author Response
Reviewer 1
Q1; In the introduction section, please clarifies abbreviation in advance.
Reply 1; There was an abbreviation MVI in the introduction. It had been corrected. Actually, the first appearance of MVI was in the Abstract. Besides, the full spelling of all abbreviations have been mentioned in the 2nd paragraph of Method.
- Q2: Please clarifies the number of patients which are involed in study
Replay 2; The total number involved in the study was descripted in the first sentence of the “Results”.
- Q3: Please describe the inclusion criteria and exclusion criteria of included patients.
Replay 3; The inclusion criteria and exclusion criteria of included patients were descripted in the first paragraph of “Study design”.
Q4: Please in the reporting of experimental measurements and of derived quantities, use internationally approved nomenclature, symbols, units, and standards.
Reply 4: Concerning using “internationally” approved nomenclature, symbols, units, and standards, because the different country had their own units for laboratory test, for example, the unit for serum bilirubin was mg/dl for Taiwan and mmol/dl for Western country. We had added the units for each laboratory test items in the 2nd paragraph “Pre-operative patient variates profiles” in the text (with underline) and Tables. Thank for your mention.
- Q5:Result section: Please focus on presenting your results with specific details.
Reply 5; Main specific points were listed as the sub-title of each paragraph of the “Results” and main results in detail were in the text. In order to avoid the repeated sentences, it will be easy to catch the main focus from the text and tables or figures together. Thanks for your suggestion.
Q6; Conclusion section: It would be beneficial to rephrase the conclusion with more comprehensive sentences that effectively summarize the key findings and implications of the study. Please consider revising the conclusion to ensure a more concise and impactful summary of the research.
Reply 6: We had revised the paragraph (with underline) of Conclusion.
Q7: Comments on the Quality of English Language
Reply 7; The manuscript had been send to MDPI langrage service for English editing. The English editing revised text have been attached together and with certificate.
Reviewer 2 Report
Comments and Suggestions for Authors
Dear Authors,
Congratulations on the article!
You imagined a practical nomogram model, integrating clinical, biological characteristics, and nutritional indices, accurately predicts MVI in HCC patients. Based on risk factors, this predictive model will help surgeons estimate MVI and optimize treatment strategies to improve long-term outcomes.
The strong point of the article is the identification of a nomogram to preoperatively evaluate patients with HCC, with the aim of early identification of MVI.
The weak points of the study are: the insufficient completion of the measurement units of the parameters mentioned in the tables, as well as the insufficient completion of the legends of these tables; lack of explanation of acronyms both in the text of the article and in the tables; insufficient completion of figure legends; non-compliance with the instructions regarding the design of the Bibliography
In the bibliography, the bibliographic references must be revised following the Publisher's Instructions for authors.
You can find the information regarding the design of the article, the design of the tables and figures, as well as the bibliography on the following link: https://www.mdpi.com/authors
The bibliography model approved by the publishing house is the one presented below:
MDPI and ACS Style
Nibbio, G.; Bertoni, L.; Calzavara-Pinton, I.; Necchini, N.; Paolini, S.; Baglioni, A.; Zardini, D.; Poddighe, L.; Bulgari, V.; Lisoni, J.; et al. The Relationship between Cognitive Impairment and Violent Behavior in People Living with Schizophrenia Spectrum Disorders: A Critical Review and Treatment Considerations. Medicina 2024, 60, 1261. https://doi.org/10.3390/medicina60081261
Please rewrite the Bibliography, starting with reference no. 11.
Tables and figures require greater attention regarding acronyms (must be explained in each table and figure); the lack of measurement units of the analyzed parameters and the legends (incomplete).
I am attaching the article with the mentioned corrections!
The article can be published after careful revision!

Author Response
Reviewer 2
Q1; The strong point of the article is the identification of a nomogram to preoperatively evaluate patients with HCC, with the aim of early identification of MVI.
Reply 1: Thanks. For surgeons to estimate the probability of MVI before resection, consequently, surgical strategies and post-operative care programs can be adapted to improve the prognosis of HCC patients where possible.
Q2: The weak points of the study are: the insufficient completion of the measurement units of the parameters mentioned in the tables, as well as the insufficient completion of the legends of these tables; lack of explanation of acronyms both in the text of the article and in the tables; insufficient completion of figure legends; non-compliance with the instructions regarding the design of the Bibliography.
Reply 2:
1). We had added the units for each laboratory test items in the 2nd paragraph “Pre-operative patient variates profiles” in the text (with underline) and Tables. Thank for your mention.
2). the full spelling of all abbreviations have been mentioned in the 2nd paragraph of Method. During preparation of this manuscript, we have to avoid the possibility of repeat sentences in the text and legends of the Figure or table. We will revise again for improving the quality of the article.
Q3; In the bibliography, the bibliographic references must be revised following the Publisher's Instructions for authors.
Reply 3. Of course, we had to follow the guideline of formation of the references. Initially, the full data of references have been prepared (automatically formatted by Endnote, providing the ODI number and easy for reviewer to find the coded original article). Finally, we will reformat the references to fit the requirement of the Journal before publication carefully.
Q4. Tables and figures require greater attention regarding acronyms (must be explained in each table and figure); the lack of measurement units of the analyzed parameters and the legends (incomplete).
Reply 4. There are many laboratory test items, all their full spellings have been completely descripted in the Methods. Their measurement units of all items have been added in the text and Tables.
Q5. I am attaching the article with the mentioned corrections!
Reply 5. Extremely sorry, I can’t find this article.
Reviewer 3 Report
Comments and Suggestions for Authors
In this study, the authors seek a novel model to predict microvascular invasion of hepatocellular carcinoma. The authors established a nomogram to predict MVI for HCC patients before surgery. There are many previous papers with similar approaches and this study does not introduce novel biomarkers that are useful for prognosis prediction, so this study lacks a little of novelty. There are a few minor comments for this manuscript.
· The authors should provide data to support the potentials of the nomogram. Nomogram should be included in Figure 5 to compare the performance between the nomogram and other factors, such as AFP. Figure 5 can be merged in one image with different colors.
· The authors should proofread the manuscript carefully, especially abbreviations. Some abbreviations are not spelled out and not clear what they mean.
· The authors should discuss more about limitations in this study, rather than sample size. Any other factors or risk factors to be considered, such as ALT? Micro (+) group has significantly higher numbers of 2+3 BCLC or T size, then it is not surprising that this group has MVI. Any thought to consider to have a cohort with no significant difference in BCLC or tumor size etc.?
Comments on the Quality of English Languagecareful proofreading is required
Author Response
Reviewer 3
- Q1. The authors should provide data to support the potentials of the nomogram. Nomogram should be included in Figure 5 to compare the performance between the nomogram and other factors, such as AFP. Figure 5 can be merged in one image with different colors.
Reply 1. The nomogram was constructed based on the univariate and multivariate logistic regression analysis. These significant variates selected from Supplementary Table S1 and Table S2). Of course, every variates of the HCC patients have their roles in the tumor recurrence and prognosis, but different in their weights. In our study, the measurement’s powers were listed in the Table 3 and 4, and Figure 5. Some risk factor might be neglected due to the significance after regression analysis. That’s the reason why the risk factor was different reported from previous studies in different countries. Besides, we had to merge as one figure for figure 5. Thanks for your suggestions.
- Q2: The authors should proofread the manuscript carefully, especially abbreviations. Some abbreviations are not spelled out and not clear what they mean.
Reply 2; Because too many laboratory test items, their all full spelling will be descripted in the Methods. Their units of all items had been added in the text and Tables. We will read and check again to avoid this mistake. Thank a lot.
- Q3:The authors should discuss more about limitations in this study, rather than sample size. Any other factors or risk factors to be considered, such as ALT? Micro (+) group has significantly higher numbers of 2+3 BCLC or T size, then it is not surprising that this group has MVI. Any thought to consider to have a cohort with no significant difference in BCLC or tumor size etc.?
Reply 3. In addition to the sample size was not large enough to have a perfect prediction. We had added some sentences (under the line) in the text. Concerning significant difference in BCLC or tumor size etc, is out-off the scope of current study but we will continue to study the recurrence rate after operation based on the different variables in our future. Thank for your suggestion.
Reviewer 4 Report
Comments and Suggestions for Authors
Dear authors
The paper is very interesting and open a pathway for adjuvant treatment in case of presence of MVI - in your center, what is the procedure in case of MVI? do you perform surgery at the same way and after systemic treatment?
Comments on the Quality of English LanguageThe quality of English language is very low - you should improve it a lot.
Author Response
Reviewer 4
Q1; The paper is very interesting and open a pathway for adjuvant treatment in case of presence of MVI - in your center, what is the procedure in case of MVI? do you perform surgery at the same way and after systemic treatment?
Reply 1. This wonderful question is our main purpose to treat the HCC patients with MVI before operation. We will continue to perform prospected study. In our personal limited experiences, the recurrent time was longer if the patient received pre-operative TACE. The final results will come out next 1-2 years. Concerning this point had been mentioned in the last 2-3rd paragraph in the “Discussion”.
Q2: The quality of English language is very low - you should improve it.
Reply 2: Thanks and this manuscript had been send to MDPI langrage service for English editing. The English editing revised text have been attached together and with certificate.
Round 2
Reviewer 1 Report
Comments and Suggestions for Authors
Please ensure that the manuscript adheres to the journal's formatting guidelines, including text size and font.
Comments on the Quality of English LanguageI appreciate for revising the manscuript and addressing all the comments
Reviewer 4 Report
Comments and Suggestions for Authors
After the answers, the paper is ready to be accepted.